# Molecular basis for polysaccharide recognition and modulated ATP hydrolysis by the O antigen ABC transporter

Nicholas Spellmon [1,2], Artur Muszyński [3], Ireneusz Górniak [1,2], Jiri Vlach [3], David Hahn[3], Parastoo Azadi[3] & Jochen Zimmer [1,2] ✉

O antigens are ubiquitous protective extensions of lipopolysaccharides in the extracellular leaflet of the Gram-negative outer membrane. Following biosynthesis in the cytosol, the lipid-linked polysaccharide is transported to the periplasm by the WzmWzt ABC transporter. Often, O antigen secretion requires the chemical modification of its elongating terminus, which the transporter recognizes via a carbohydrate-binding domain (CBD). Here, using components from *A. aeolicus*, we identify the O antigen structure with methylated mannose or rhamnose as its cap. Crystal and cryo electron microscopy structures reveal how WzmWzt recognizes this cap between its carbohydrate and nucleotide-binding domains in a nucleotide-free state. ATP binding induces drastic conformational changes of its CBD, terminating interactions with the O antigen. ATPase assays and site directed mutagenesis reveal reduced hydrolytic activity upon O antigen binding, likely to facilitate polymer loading into the ABC transporter. Our results elucidate critical steps in the recognition and translocation of polysaccharides by ABC transporters.

Many Gram-negative pathogens evade host innate immune responses by depositing protective carbohydrate polymers on their cell surfaces. The chief molecular constituents of the outermost leaflet of the Gram-negative outer membrane (OM) are lipopolysaccharides (LPS), which are glycolipids composed of a structurally conserved lipid-A and core oligosaccharide (hereafter lipid-A core), as well as O antigen polysaccharides[1]. O antigens are hypervariable with serotype-specific repeat units responsible for coating and expanding the bacterial envelope. Next to serving as an important defense layer, O antigens also form diffusion barriers to many solutes that contribute to antimicrobial resistance[2,3].

The ABC transporter-dependent pathway is one system used to synthesize and export O antigens from the inner membrane (IM)[4]. Here, the O antigen polymer is first synthesized in the cytoplasm as a glycolipid intermediate on an undecaprenyl-pyrophosphate anchor. The completed glycoconjugate is exported by the WzmWzt ABC

transporter to the periplasm where the O antigen is transferred to the lipid-A core to form a mature LPS molecule that is shuttled to the outer leaflet of the OM[5]. Wzm forms the ABC transporter's membrane-embedded permease domain and Wzt contains the classical cytosolic nucleotide-binding domain (NBD).

The ABC transporter-dependent pathway is further categorized into two unique systems defined by the reliance on a terminal O antigen modification or lack-thereof. This 'cap' is attached to the growing non-reducing end of the O antigen polymer. Caps can either be a chemical modification of the terminal sugar, including methylation and (methyl)-phosphorylation, or the attachment of a unique sugar unit, such as 3-deoxy-D-manno-oct-2-ulosonic acid (Kdo)[5,6].

Cap attachment ceases O antigen elongation and certifies the polymer for export by the corresponding WzmWzt transporter containing a CBD recognizing the cap[6–8]. The CBD is essential for capped O antigen export in vivo as transporters lacking the domain or carrying

[1]Howard Hughes Medical Institute, University of Virginia School of Medicine, Charlottesville, VA, USA. [2]Department of Molecular Physiology and Biological Physics, University of Virginia School of Medicine, Charlottesville, VA 22908, USA. [3]Complex Carbohydrate Research Center, University of Georgia, Athens, GA 30602, USA. ✉e-mail: jz3x@virginia.edu

CBDs with compromised cap binding activities fail to export their lipid-linked substrates[9].

The CBD is fused in tandem to the C-terminal end of the conserved NBD of Wzt. Crystal structures of isolated CBDs from different species revealed dimeric structures of two jelly-roll domains stabilized by mutual exchange of their C-terminal β-strands[9–11]. In vitro binding studies using *E. coli* O9a and *R. terrigena* LPS indicated cap binding to a moderately conserved concave β-sheet surface of each dimer subunit.

Although the CBD is essential for O antigen secretion, individual expression of the CBD with CBD-truncated WzmWzt restores transport activity in vivo, suggesting that the domains can function in trans[9]. It is currently unknown how the CBD recognizes its substrate and how this interaction enables O antigen secretion.

Structural analyses of the WzmWzt transporter from *Aquifex aeolicus* provided the first insights into polysaccharide transporting ABC transporters. WzmWzt is a type-V ABC transporter with Wzm's transmembrane (TM) topology resembling the wall teichoic acid TarGH and eukaryotic lipid-ABCG5/8 transporters[12–14]. Notably, crystal structures of CBD-truncated WzmWzt in nucleotide-free and ATP-bound states revealed channel-forming architectures consistent with conformations during polysaccharide translocation[15,16]. A cryo-electron microscopy (cryo-EM) analysis of full-length ATP-bound WzmWzt revealed the closing of the TM channel near its periplasmic exit and, strikingly, a highly asymmetric positioning of the CBD dimer, relative to the core ABC transporter[17]. The CBD dimer interacts with only one NBD by straddling a conserved 'hinge helix' at the C-terminus of the NBD sub-domain via loops protruding from each CBD protomer. *A. aeolicus* WzmWzt is well suited for structure-function analyses; however, knowledge on the structure and capping of the corresponding O antigen polymer and its interactions with the transporter are required to mechanistically delineate O antigen recognition and transport.

In this study, we determine the structure of purified native *A. aeolicus* O antigen and identified methylated rhamnosyl and mannosyl units as the O antigen caps. A crystal structure of the isolated Wzt-CBD bound to 3-methyl mannose provides atomistic insights into cap recognition by the CBD. A cryo-EM structure of full-length WzmWzt in a nucleotide-free conformation reveals a dramatic repositioning of the CBD dimer relative to Wzt's NBD sub-domain, compared to the ATP-bound conformation. Further, O antigen-bound WzmWzt structures identify how the NBD and CBD sub-domains cooperate in O antigen binding. Lastly, in vitro O antigen and cap binding experiments demonstrate modulation of ATPase activity upon O antigen interaction, likely necessary for polymer translocation. Combined, our data elucidate a model for O antigen recognition and translocation by CBD-dependent O antigen transporters.

## Results

### *Aquifex aeolicus* synthesizes a heterogeneous rhamnose-mannose O antigen

O antigens are structurally diverse with more than 60 mono-saccharides identified to date[18]. Adding to the structural diversity, the polymers not only differ in glycosyl composition, but also in the stereochemistry of the glycosidic linkages and non-carbohydrate modifications. A previous composition analysis of *A. aeolicus* LPS extracts revealed a predominance of rhamnosyl and mannosyl units, similar to several *E. coli* O antigen compositions[19]. To aid mechanistic studies, we determined the composition, structure, and capping of the *A. aeolicus* VF5 O antigen, isolated from in vivo produced LPS.

Polyacrylamide gel electrophoresis of intact *A. aeolicus* LPS separates high and low molecular weight species (HMW and LMW, respectively) (Fig. 1a). The LMW fraction likely corresponds to the lipid A-core based on gel co-migration with a *Salmonella enterica* LPS reference. Most of the LPS, however, resolves in a HMW ladder likely representing LPS consisting of the lipid-A core substituted with long hetero-polymeric O antigens (so-called smooth LPS).

A glycosyl composition analysis of the intact LPS revealed about 73% rhamnose (Rha) and 12% mannose (Man) and trace amounts of *O*-methylated Rha and Man (discussed in detail in the following section). Additional identified glycosyl residues include galactose, glucose, and Kdo, presumed to originate from core oligosaccharides and/or the linker region connecting to the O antigen (Supplementary Table 1). To perform a detailed structural characterization of the O antigen, the polysaccharide portion of the LPS (O antigen together with the attached core) was released from lipid A by mild acid hydrolysis. Size exclusion chromatography of the obtained polymers revealed two major populations, in agreement with LPS' separation into LMW and HMW fractions by PAGE (Fig. 1a, b). Glycosyl linkage analysis of the O polysaccharide in the HMW fraction revealed 3-substituted Rha (47.0%), 2-substituted Rha (23.6%), 2-substituted Man (18.4%), and terminal Rha (5.9%), with all sugars in the pyranose (p) configuration. In addition, gas chromatography-mass spectrometry (GC-MS) of trimethylsilylated (+)−2-butyl glycosides derived from the released monosaccharides established the D absolute configuration of the Rha and Man units.

Nuclear magnetic resonance (NMR) spectroscopy provided further insights into the O antigen structure. The $^1$H and $^1$H,$^{13}$C-HSQC NMR spectra (Fig. 1c), contained several major anomeric signals that were accompanied by a set of low-intensity signals. Analysis of the 1D $^1$H and a set of 2D NMR experiments (COSY, TOCSY, NOESY, HSQC, HMBC, and HSQC-TOCSY) provided complete assignments of the $^1$H and $^{13}$C resonances of the major O antigen residues (Supplementary Fig. 1, Supplementary Table 2 and Supplementary Methods).

Despite the apparent simplicity of the $^1$H and HSQC spectra, the analysis revealed the presence of eight non-equivalent residues that form the O antigen polysaccharide. The polymer consists of two 2-substituted α-D-Man*p* residues [→2)-α-D-Man*p*(1→; residues Aa and Ab], two 2-substituted α-D-Rha*p* residues [→2)-α-D-Rha*p*(1→; residues Ba and Bb], and four 3-substituted β-D-Rha*p* residues [→3)-β-D-Rha*p*(1→; residues Ca, Cb, Da and Db] (Supplementary Figs. 1a, b and Supplementary Table 2). Residues A are linked to residues C, while residues B are linked to residues D, thus forming →2)-α-D-Man*p*(1→3)-β-D-Rha*p*(1 and →2)-α-D-Rha*p*(1→3)-β-D-Rha*p*(1 pairs in the molar ratio of (AC)₂(BD)₃, (Fig. 1c, d). The NOESY data (Supplementary Fig. 1b) showed that the AC and BD pairs could be linked interchangeably, forming all four possible combinations (ACAC, ACBD, BDAC and BDBD) and resulting in the appearance of the eight non-equivalent residues (Fig. 1d). Our NMR data do not allow determining whether a defined repeat unit exists or whether the sequence of the AC and BD pairs is semi-random.

### Methylated rhamnose and mannose cap the *Aquifex aeolicus* O antigen

As part of the linkage and composition analyses described above, the EI-MS analysis of alditol acetates derived from the O antigen revealed that approx. 1.6% of the detected Rha units is 3-*O*-methylated (3-*O*-Me-Rha), and, similarly, that approx. 4.8% of the detected Man residues is 3-*O*-methylated (3-*O*-Me-Man) (Supplementary Fig. 1e, f). Likewise, the NMR spectra of the O antigen showed that two of the minor anomeric signals belong to terminal 3-*O*-Me-α-Man*p*(1→ (residue F) and 3-*O*-Me-α-Rha*p*(1→ (residue G) (Supplementary Fig. 1d) and that these terminal residues are linked to C-like and D-like glycosyl residues, respectively. Based on the integrals of their HSQC anomeric signals, the terminal residues are approximately equimolar. Other minor anomeric peaks (E, H, I and J in Supplementary Fig. 1) likely correspond to residues of the linker region. Combined, our linkage and NMR analyses show that methylated Rha or Man cap the Rha/Man-rich *A. aeolicus* O antigen at its non-reducing terminus.

### *Aquifex aeolicus* produces a large O antigen polymer

We next analyzed the average length of the *A. aeolicus* O antigen. First, we determined molecular weights of the acid-released polysaccharide

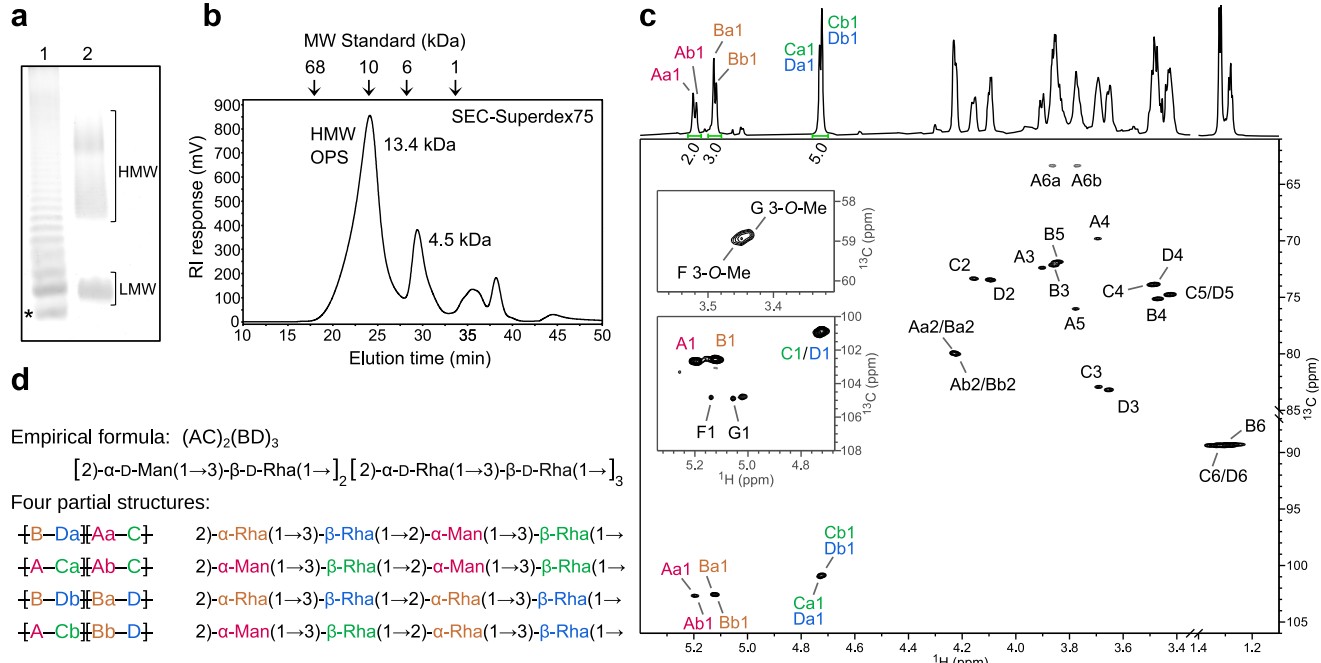

**Fig. 1 | Structure of the *A. aeolicus* O antigen. a** Representative SDS-PAGE of *A. aeolicus* VF5 LPS. Lane 1: *Salmonella enterica* LPS; Lane 2: *A. aeolicus* VF5 LPS. In lane 1, the LPS band with the smallest molecular weight is the lipid A-core (*). The subsequent bands above the lipid A-core is lipid A-core with one repeat unit consecutively added after each band. An uncropped gel image is shown as Source Data. **b** Size exclusion of purified O antigen. The poly-saccharide elutes at the first peak with an average molecular weight of 13.4 kDa. **c** Partial ¹H and ¹H,¹³C-HSQC spectra of the released HMW O poly-saccharide chain with signal assignments. Anomeric peak areas are shown below the ¹H spectrum. The two insets of HSQC spectrum at lower contour levels show the anomeric and methyl group signals of the terminal residues F and G. The signals of rhamnose methyl groups (B6, C6 and D6) are aliased along F1 and their actual ¹³C chemical shifts are lower by 70 ppm. **d** Structural motifs present in *A. aeolicus* LPS O antigen. The partial structures illustrate the origin of the a/b non-equivalency in the major residues A, B, C and D. No regular repeating unit could be determined, so a semi-regular repeating structure is proposed.

fractions by size exclusion chromatography, using known carbohydrate standards (Fig. 1b). Because the HMW fraction (with an estimated MW of ~13.4 kDa) contains O antigens attached to the core oligosaccharides and the LMW species (with an estimated MW of ~4.5 kDa) represents predominantly core oligosaccharides (Supplementary Table 1), an O antigen MW of about 8.9 kDa can be deduced from the mass difference. Second, the MW of this O antigen was corroborated by ¹H NMR experiments based on the areas of anomeric signals of residues A–D and the terminal residue G (Supplementary Fig. 1d). Accordingly, NMR analysis indicates an average O antigen length of ~63 residues, corresponding to a MW of ~9.4 kDa for an O antigen containing 80% rhamnosyl and 20% mannosyl units, in agreement with the size estimate derived from size exclusion chromatography.

## *Aquifex aeolicus* Wzt-CBD specifically recognizes 3-*O*-methyl-D-mannose

Wzt's CBD is predicted to bind the O antigen cap, thus the identified 3-*O*-methyl mannosyl and rhamnosyl units are expected to interact with the CBD. To test this hypothesis, we performed isothermal titration calorimetry with the isolated Wzt-CBD and 3-*O*-Me-D-Man. Indeed, each CBD monomer binds the methylated monosaccharide with about 234 μM affinity and at a 1:1 molar ratio (Fig. 2a and Supplementary Fig. 2). No binding is observed to unmodified D-mannose, suggesting that the CBD specifically recognizes the sugar's O3 substitution. These experiments also confirm the D-configuration of the 3-*O*-Me-Man cap.

To delineate how the CBD recognizes the cap, we crystallized the isolated *A. aeolicus* CBD in the presence of 3-*O*-Me-D-Man and solved the molecular structure at 1.6 Å resolution by molecular replacement (Supplementary Table 4 and Methods). The overall topology of the dimeric CBD exhibits a β-sandwich fold, nearly identical to our previously reported apo structure (PDB: 6O14, RMSD = 0.34 Å)[10] (Fig. 2b).

Following modeling and refining the protein component only, an unbiased FoFc difference maprevealed excellent electron density for the bound 3-*O*-Me-D-Man ligand (Fig. 2c). The monosaccharide binds to each CBD protomer on their flat β-sheet surfaces, formed by β-strands 3, 4, 6 and 7 (Fig. 2b–e).

The mannose ring is oriented by a delicate hydrogen bond network (Fig. 2e). Specific contacts are formed between the ring oxygen and the δ² amide from Asn313, the C2 hydroxyl and the ε²-nitrogen from His350, as well as the C4 hydroxyl and the carbonyl backbone from Ile299. The C6 hydroxyl group, which is absent in Rha, only undergoes water-mediated interactions with the carbonyl of Met353. Compared to CBD's apo conformation, these interactions position its loop connecting β-strands 6 and 7 (residues 352-356, hereafter referred to as 'trap-loop') towards the ligand-binding site, thereby trapping the O antigen cap. The 3-*O*-methyl moiety fits into a small hydrophobic pocket formed by Leu300, Thr346, Ala348 and Trp362 (Fig. 2d, e). The indole ring from Trp362 stacks against the methyl cap through CH-π interactions. Altogether, our data demonstrate that *A. aeolicus* CBD specifically recognizes the methyl-cap on the non-reducing end sugar of the O antigen via interactions that would also be consistent with 3-*O*-methyl-Rha forming the cap.

## The CBD dimer undergoes large, nucleotide-induced, rigid body movements

Next, we used single particle cryo-EM to identify conformational changes of the CBD during the ATP hydrolysis cycle of the O antigen ABC transporter. We determined the structure of full-length WzmWzt reconstituted into a lipid nanodisc in nucleotide-free (apo) and ADP-bound states (Fig. 3a, Supplementary Fig. 3 and Supplementary Table 4). The overall conformation of the transporter in the apo and ADP-bound states is essentially same, so we selected the ADP-bound

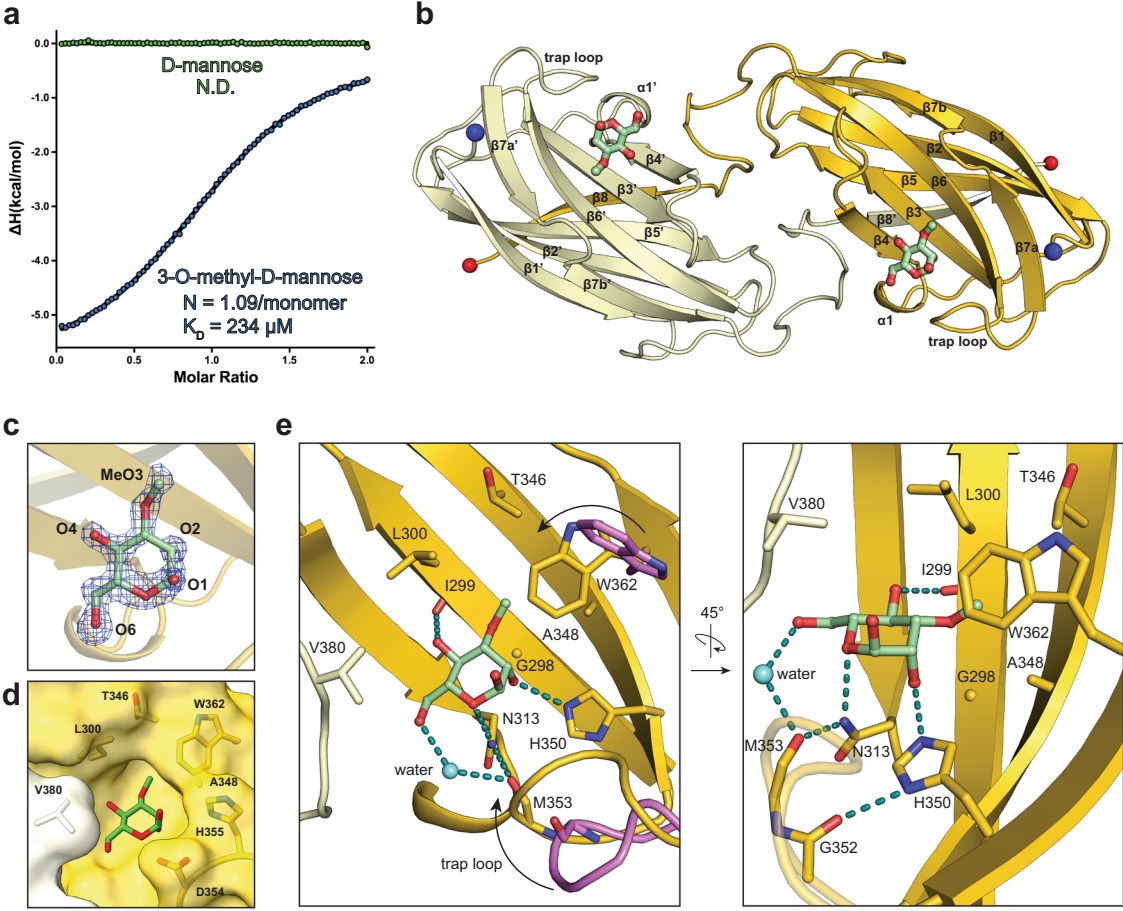

**Fig. 2 | Molecular structure of Wzt-CBD bound to 3-O-methyl-ᴅ-mannose.**
**a** Isotherm plot of Wzt-CBM against 3-O-methyl-ᴅ-mannose (blue) and ᴅ-mannose (green) Raw data is provided as Source Data. **b** Overall structure of the Wzt-CBD dimer. Two monomers are colored separately as yellow and vanilla. 3-O-methyl-ᴅ-mannose is shown as green sticks. N- and C-terminal ends are shown in blue and red spheres, respectively. **c** Modeled 3-O-methyl-ᴅ-mannose into an unbiased $F_OF_C$ map. Oxygen atoms are labeled by atom number. Mesh map was contoured at 3σ.

**d** Surface representation of key residues that form the cap sugar-binding pocket. **e** Two orientations of key interactions between 3-O-methyl-ᴅ-mannose and Wzt-CBM. Trap loop and Trp362 from apo Wzt-CBM (PDB ID: 6O14) are colored in violet. The cap-induced movement of the trap loop and Trp362 is indicated with an arrow. Water is represented as a light blue sphere. Hydrogen bonds are shown as dashed lines colored in teal.

model for analysis. Under these conditions, the transporter adopts a teepee-like conformation in which the NBDs are separated. In this structure, NBD opening significantly exceeds the previously described apo conformation of the CBD-truncated WzmWzt transporter, (Fig. 3a and Supplementary Fig. 4a)[15]. The open conformation of the full-length transporter remains unaltered in the presence of ADP:Mg²⁺, which binds to the canonical nucleotide binding site on each Wzt subunit (Supplementary Fig. 4b). However, compared to the transporter's ATP-bound closed conformation[17], the CBD dimer undergoes major rigid-body movements away from the cytosolic water-lipid interface (Fig. 3a). Accordingly, extensive structural rearrangements are observed at the interface of Wzt's NBD and CBD sub-domains.

Wzt's NBD contains three α-helices at the interface with the CBD dimer. Starting at the C-terminal end of the NBD, these include the hinge helix (residues 226-240) that mediates the interaction with the CBD dimer in the ATP-bound conformation, a short helix following the conserved H-loop (residues 201 to 207, hereafter referred to as 'H-loop-helix'), and an elongated helix at the interface with NBD's helical sub-domain (residues 174 to 189) (Fig. 3a).

While the overall organization of the CBD dimer (formed from Wzt subunits 1 and 2, denoted CBD1 and 2) remains unchanged relative to its isolated conformation or as part of ATP-bound full-length WzmWzt (Supplementary Fig. 4c), the dimer translates and rotates along its long axis, relative to the NBD it is interacting with (from Wzt

subunit 1, denoted NBD1). Therefore, it no longer straddles the hinge helix, and instead interacts with the neighboring H-loop-helix (Fig. 3a). This transition moves CBD1 (referred to as CBD1-a, 'a': active) away from the membrane, while CBD2, connected to NBD2 of the other protomer (CBD2-i, 'i': inactive), transitions towards the membrane interface (Supplementary Movie 1).

In this position, the same conserved CBD loop that straddles the hinge helix in the closed ATP-bound conformation (residues 302-308) now contacts the H-loop-helix (Fig. 3a). In particular, CBD1-a creates a hydrogen bond network via Arg302 and Asp308 with Asn202 and Lys205 of the H-loop-helix as well as Gln230 and Tyr233 of the hinge helix. Further, Phe305 of CBD1-a stacks against Ile206 of the H-loop-helix. In the CBD2-i protomer, this residue packs into a hydrophobic pocket formed by Ala203, Ile206, and Leu207 of the H-loop-helix as well as Phe181 of the NBD's elongated helix. In the apo conformation, the transporter's NBD and CBD sub-domains share a surface interface of approx. 2706 Å², compared to 2443 Å² in the ATP-bound state (PDBePISA).

Modeling the CBD dimer in its new position onto the transporter in the closed ATP-bound conformation reveals major clashes between the CBD and NBD sub-domains. This suggests that the CBD position observed in the apo conformation is energetically favored, yet nucleotide-induced NBD closure displaces the CBD dimer to its peripheral position where it interacts with the hinge helix (Fig. 3b).

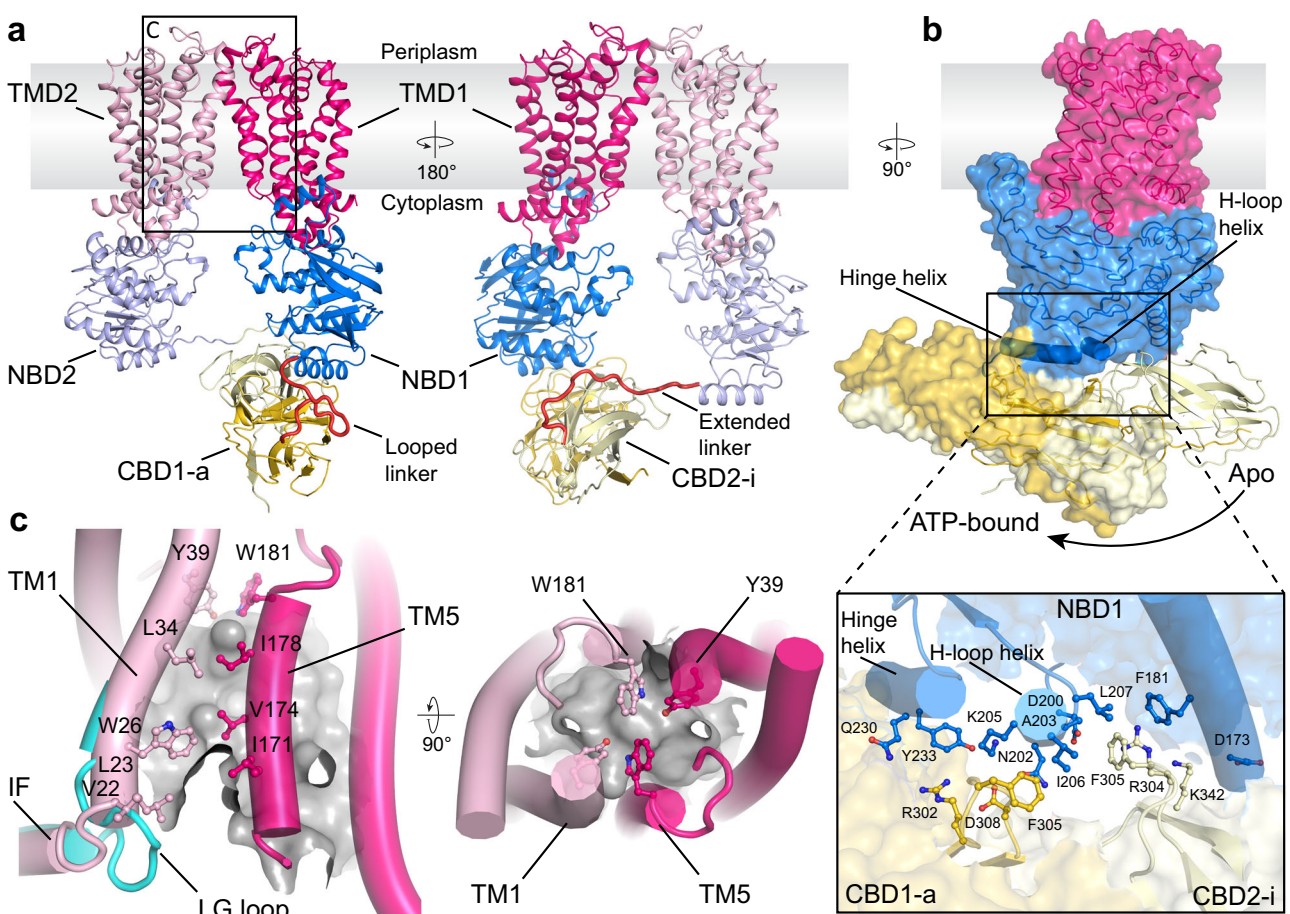

**Fig. 3 | Cryo-EM structure of the ADP-bound WzmWzt O antigen transporter.** **a** Overview of the WzmWzt structure. Loops connecting the NBD with the CBD are colored red. The boxed region is enlarged in panel **c**. TMD: Transmembrane domain. **b** CBD position in the transporter's ADP and ATP-bound conformations. The ATP-bound conformation is shown as a semi-transparent surface. The panel presents interactions between the NBD1 and CBD (ADP-bound). **c** The LG loop moves away from the channel entrance upon NBD opening. The cyan cartoon represents the previously reported IF-TM1 connection observed in the ATP-bound WzmWzt (PDB ID: 7K2T). The TM channel is shown as a gray surface (calculated in HOLLOW[35]) and aromatic belt residues are shown as sticks.

Further, the linker regions connecting the NBD and CBD sub-domains (residues 241–249) are structured differently, compared to the ATP-bound closed conformation (Fig. 3a). The linker connecting CBD1-a with the preceding NBD1 bends into a hairpin-like loop between the NBD's hinge helix and the N-terminal β-strand of CBD1-a. The linker connecting CBD2-i to the opposing NBD2 is an extended conformation that spans about 24 Å.

**An asymmetric TM organization**

In the apo conformation, the Wzm subunits form a membrane-spanning channel that is closed towards the periplasm by the aromatic ring residues, similar to the hydrophobic seal observed in the ATP-bound conformation[17] (Fig. 3c). The lipid-linked O antigen substrate likely enters the transporter with its head group first at the cytosolic channel entrance formed at the interface of the Wzm subunits, between TM helices 1 and 5 of opposing subunits. The wide spreading of Wzt's NBDs imposes an opening of this cytosolic gate.

Rearrangements within the TM region are asymmetric, with a slightly wider opening of the TM1/TM5 seam above the CBD1-a domain. At this portal, the LG-loop connecting Wzm's N-terminal interface helix with TM helix 1 flips away from the channel axis, relative to its conformation in the closed ATP-bound state[16] (Fig. 3c and Supplementary Fig. 4e). The partially open channel entrance stabilizes several hydrophobic non-proteinaceous molecules, likely lipids, at the TM1/TM5 interface. In particular, Val22, Leu23, Trp26, and Leu34 of

TM1 and Ile171, Val174, and Ile178 of TM5 in the opposing Wzm subunit create a lipid-stabilizing hydrophobic pocket. Additional ordered lipid molecules are observed surrounding the periplasmic termini of the TM helices (Supplementary Fig. 4d).

**Wzt's CBD dimer and one NBD sub-domain create a single O antigen-binding pocket**

To identify how WzmWzt interacts with the O antigen, we determined cryo-EM structures of the nucleotide free full-length transporter bound to either 3-O-Me-Man or the in vivo synthesized *A. aeolicus* O antigen (Fig. 4, Supplementary Fig. 5 and Supplementary Table 5).

First, 3-O-Me-Man was added to the purified WzmWzt transporter preceding cryo grid preparation. The obtained structure at an estimated resolution of 3.7 Å is identical to the above described apo or ADP-bound conformation. However, additional density is evident at the cap-binding site of each CBD domain (Fig. 4b).

Second, two different methods were used to determine the structure of WzmWzt tethered to the *A. aeolicus* O antigen. We supplied purified O antigen to transporter-containing nanodiscs prior to cryo grid preparation, and we co-reconstituted WzmWzt together with *A. aeolicus* LPS into nanodiscs. Both strategies resulted in EM maps exhibiting strong extra bilobal density only at CBD1-a (Fig. 4a and Supplementary Fig. 5a). Accordingly, in this domain, the trap-loop is tilted inwards to coordinate the ligand, as observed in the cap-bound crystal structure (Fig. 2e). In the opposing CBD2-i, however, the cap-binding site is unoccupied (Fig. 4a).

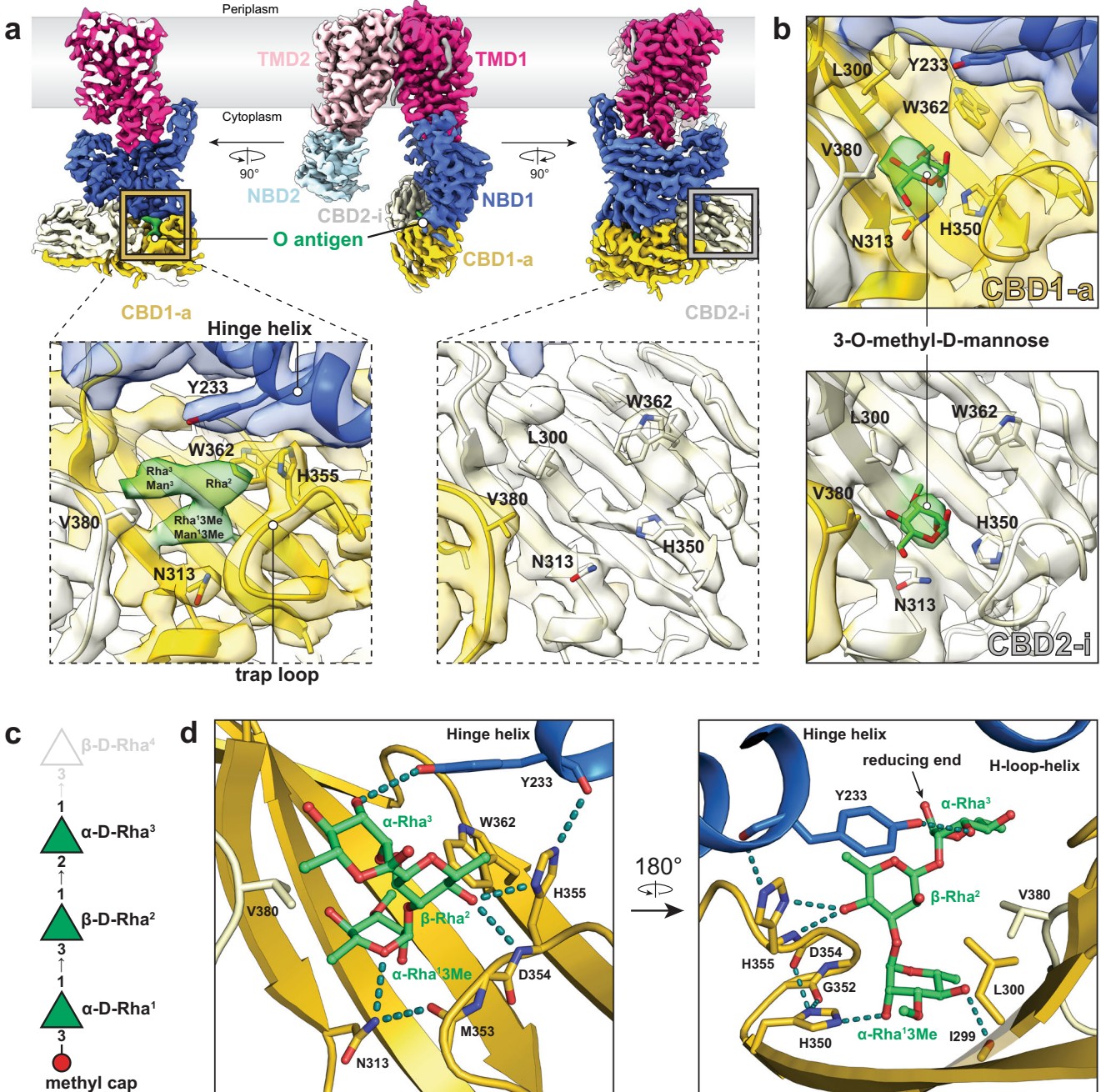

**Fig. 4 | Cryo-EM structure of WzmWzt in complex with *A. aeolicus* O antigen or 3-*O*-methyl-ᴅ-mannose. a** Coulombic potential map of WzmWzt in complex with purified *A. aeolicus* O antigen. Three different side orientations are shown. Boxed panels show local density within the cap-binding site for CBD1-a and CBD2-i. Density for O antigen is colored in green; contour level: 8.9σ. **b** Panels show local density within the cap-binding site for CBD1-a and CBD2-i from the 3-*O*-methyl-ᴅ-mannose bound WzmWzt map. Density for 3-*O*-methyl-ᴅ-mannose is colored in green.; contour level: 7.2σ (CBD2-i site) and 5.4σ (CBD1-a site). **c** Model substrate

used to dock into the O antigen density. ᴅ-rhamnose is represented as green triangles. Reducing and non-reducing ends are on the top and bottom, respectively. The superscript (Rhaⁿ) in each sugar denotes its position in the O antigen sequence starting from the non-reducing end. β-ᴅ-Rha⁴ (gray) was not included in the OPS model, thus the reducing end of α-ᴅ-Rha³ was left unlinked. **d** Molecular interaction between Wzt and *A. aeolicus* O antigen. Two different orientations are shown. The carbon backbone of the O antigen is colored in green. Hydrogen bonds are shown as dashed lines colored in teal.

The density observed at the CBD1-a ligand-binding site readily accommodates three glycosyl units. We modeled the substrate based on the expected composition and linkage of the *A. aeolicus* O antigen with 3-*O*-Me-Rha at the non-reducing end (Figs. 4c, d and 1d). Modeling the substrate was guided by (1) the position of the capping sugar based on our crystal structure and (2) the α-glycosidic linkage connecting the first and second glycosyl units, which presents a kink in the polymer. Since the map quality for the third sugar unit is weak (Supplementary Fig. 5a), the modeled orientation of ᴅ-Rha³ is only approximate.

The interactions of Wzt's CBD dimer and NBD1 with the O antigen exhibit a network of hydrogen bonds and van der Waals contacts. The 3-*O*-Me-Rha¹ cap (which could also be 3-*O*-Me-Man) is coordinated as described above for the 3-*O*-Me-Man bound CBD (Fig. 2). CBD1-a's His350 connects the capping sugar with the trap-loop via hydrogen bonds between the cap's C2 hydroxyl and the carbonyl backbone of the loop's Asp354 (Fig. 4d).

The following glycosyl unit, β-ᴅ-Rha², is stabilized by the side chains of Tyr233 of NDB1's hinge helix, His355 of CBD1-a, as well as

Val380 of CBD2-i. The last resolved sugar unit, α-D-Rha[3], contacts the Val380 side chain and its C2 hydroxyl group is in hydrogen bond distance to the hydroxyl of NBD1's Tyr233.

His355 serves as an important nexus to stabilize the interaction of CBD1-a with the O antigen and the NBD1 sub-domain (Fig. 4d). The side chain forms hydrogen bonds with the C4 hydroxyl of the second rhamnosyl unit as well as the backbone carbonyl of Tyr233 in NBD1's hinge helix, thereby locking the O antigen in a deep binding pocket at the interface of Wzt1's sub-domains.

The side chain of NBD1 Tyr233 appears to be key for contacting the O antigen. The arrangement of the CBD dimer and NBD1 appears necessary for Tyr233 to contact a single O antigen chain. CBD2-i likely does not possess an O antigen since its O antigen binding pocket is segregated away from Tyr233 of NBD2.

## ATP binding and NBD closure destabilizes CBD's interaction with the O antigen

The above-described O antigen-binding pocket is dramatically altered in the transporter's closed ATP-bound conformation. This is due to the profound rotation and translation of the CBD dimer relative to the interacting NBD (Fig. 3b). Therefore, we investigated whether the transporter continues to interact with the O antigen in the ATP-bound state or whether alternative conformations are stabilized.

To this end, the purified O antigen was added to nanodisc-reconstituted wild type WzmWzt to obtain the polymer-bound structure as described above. After an incubation period of 4 min on ice, ATP was added, followed by an additional incubation for 1 h at 4 °C prior to cryo grid preparation. Cryo-EM analyses of the obtained dataset revealed at least two distinct transporter conformations, corresponding to its NBD open and closed states (Fig. 5a, b and Supplementary Fig. 6a).

The map of the open conformation reveals one ADP:Mg$^{2+}$ molecule at one NBD and, importantly, the bound O antigen chain at the interface of the CBD1-a and corresponding NBD1, as described above (Figs. 5c and 4). In the ATP-bound closed conformation, however, the CBD dimer is located at the previously described alternative position straddling the hinge helix (Fig. 5d). This map lacks any features indicating the presence of a stably bound O antigen at the identified binding site of CBD1-a, consistent with the disruption of its binding pocket upon CBD displacement (Fig. 5c, d). Owing to the displacement, polysaccharide-coordinating residues within CBD1-a (for example His350) and NBD's hinge helix (i.e., Tyr233) are now separated by about 30 Å.

## Site-directed mutagenesis of cap binding residues disrupts O antigen binding and substrate-induced ATPase activity

Our structural analyses identified several residues of the O antigen-binding pocket likely critical for polysaccharide binding. In vitro LPS pull-down experiments with purified CBD constructs confirm their importance for ligand interaction (Fig. 6a). Wild type CBD dimers immobilized on metal affinity chromatography resin bind purified *A. aeolicus* LPS, revealing a characteristic banding pattern upon silver staining of the eluted fractions. Replacing Trp362, which lids the 3-*O*-methyl binding pocket, with leucine essentially abolishes LPS binding. Similarly, substituting Val380, provided by the neighboring CBD protomer to coordinate the cap's C6 position, with glycine also nearly abrogates in vitro LPS binding. Replacing His355 with alanine, which bridges the C4 hydroxyl of the polymer's second rhamnosyl unit with the backbone carbonyl of Tyr233 of the NBD's hinge helix in the full-length transporter, had no detectable effect on LPS binding (Fig. 6a). This suggests that this interaction is not critical in the context of an isolated CBD dimer.

The same mutations in the micelle-embedded full-length WzmWzt show a less dramatic reduction in binding capability (Supplementary Fig. 7a, b). We observe significant non-specific LPS binding,

likely due to interactions of the lipid moiety with the transporter and/ or the surrounding detergent micelle. Similarly, high levels of non-specific LPS binding are observed with a CBD-truncated transporter as well as when purified Kdo$_2$-lipid-A (Avanti Lipids) instead of *A. aeolicus* LPS is used (Supplementary Fig. 7c). In addition, LPS pull-down experiments with nanodisc-reconstituted WzmWzt (as used for our cryo-EM analyses), failed to show LPS binding, in contrast to our cryo-EM analyses. This could be due to lower packing density of the immobilized transporter on the metal affinity beads, thereby decreasing avidity.

We next monitored WzmWzt's ATPase activity in the presence of *A. aeolicus* LPS or the released O antigen polysaccharide. First, comparing the transporter's ATP hydrolysis rates in a detergent micelle and lipid nanodisc environment reveals about 25% ATPase activity in a DDM micelle (*in surfo*), relative to nanodisc-reconstituted transporter (Fig. 6b). Second, increasing concentrations of purified LPS stimulate the transporter's hydrolytic activity *in surfo* to roughly the levels observed in nanodiscs, while similar concentrations of the released O antigen polysaccharide (without the lipid moiety) only minimally affect catalysis (Fig. 6c). Third, in nanodiscs, *A. aeolicus* LPS as well as the released O antigen reduce WzmWzt's ATPase activity by over 50%, compared to its activity in the absence of LPS (Fig. 6d). We conclude that the stimulatory effect exerted by LPS on detergent-solubilized WzmWzt reports on O antigen binding, yet is likely due to secondary effects induced by LPS' lipid groups, perhaps by modulating micelle properties.

This interpretation is confirmed by mutations in the cap-binding pocket compromising O antigen binding (Fig. 6b). *In surfo*, WzmWzt's catalytic activity in the presence of saturating LPS concentrations drops to basal levels (i.e., absence of LPS) upon replacing Tyr233 or His355 with Ala, or Trp362 with Leu. Only the V380G mutant exhibits a remaining ~2-fold increase in ATPase activity in the presence of LPS, stressing its peripheral interaction with the capping O antigen sugar (Fig. 2e). We note that the His355 to Ala substitution did not significantly affect LPS binding by the immobilized CBD, yet it impacts stimulation of hydrolytic activity in the full-length transporter. These differences may arise from increased avidity of the immobilized CBDs.

In a nanodisc-reconstituted system, ATPase inhibition is only observed for the wild type enzyme. LPS addition to the W362L mutant, which abolishes LPS binding of the isolated CBD dimer, does not impact hydrolytic activity (Fig. 6a, b).

Reactions in the presence of 3-*O*-methyl-D-Man instead of O antigen reveal no significant effect on WzmWzt's ATPase activity (Fig. 6c, d). At non-physiologically high concentrations, however, 3-*O*-Me-D-Man (and not D-Man) increases WzmWzt's hydrolytic activity by about 50%. Combined, our experiments suggest that the coordination of the terminal O antigen trisaccharide at Wzt's CBD/NBD interface reduces the transporter's catalytic activity under physiological conditions.

## WzmWzt forms a continuous TM channel

*In surfo*, the LPS-induced increase in catalytic activity of WzmWzt prompted us to determine its structure in a DDM micelle environment. We were unable to obtain an O antigen bound structure due to limiting sample quality, but we did resolve the structure of the full-length WzmWzt in a nucleotide-free state. Although at a lower estimated resolution of about 4 Å, the cryo-EM map of nucleotide-free WzmWzt reveals a unique architecture (Fig. 6e, Supplementary Fig. 6b and Supplementary Table 5). The structure resembles the previously determined apo conformation of the CBD-truncated construct[15] (Fig. 6f). Here, Wzt's NBDs are less separated compared to the nanodisc-reconstituted apo state (Fig. 3a) yet provide sufficient room for nucleotide diffusion (Supplementary Fig. 4a). Further, the Wzm TM subunits and in particular their aromatic belt residues dilate to form a TM channel,

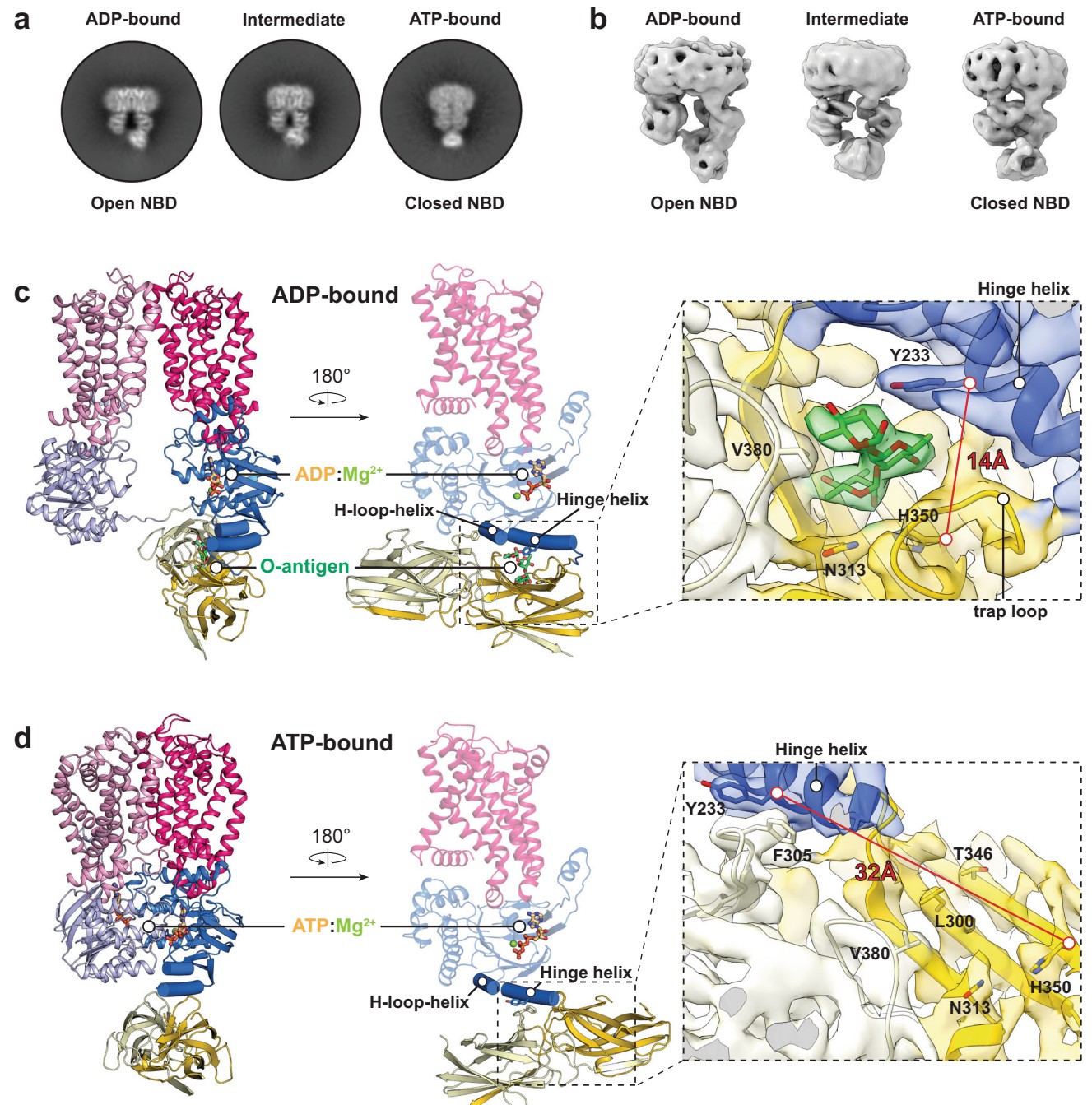

**Fig. 5 | ATP binding displaces the O antigen from the CBD. a, b** Representative 2D and 3D classes of WzmWzt conformational states identified during ATP hydrolysis. **c** Cartoon model of WzmWzt bound to ADP:Mg$^{2+}$ and O antigen. Domains are colored as shown in Fig. 3. The O antigen density is colored green. O antigen and ADP are presented as sticks and Mg$^{2+}$ as a lime sphere. Zoom window: Local map of the O antigen binding site for CBD1-a in the ADP-bound map contoured at 7.2σ. **d** Cartoon model of WzmWzt bound to ATP:Mg$^{2+}$. ATP is presented as sticks and Mg$^{2+}$ as a lime sphere. Zoom window: Local map of the O antigen binding site for CBD1-a in the ATP-bound map contoured at 7.7σ.

as previously reported for the ATP-bound and apo states of the CBD-truncated WzmWzt[15,16].

The TM channel is continuous from the periplasmic to the cyto-solic side of the membrane (Fig. 6e). The channel is about 40 Å in length, which suffices to accommodate an octasaccharide in an extended conformation. In contrast to the previously reported channel-forming apo conformation, TM helix 1 at the interface of the Wzm dimer is shifted by about 10 Å towards the channel axis, thereby creating profound lateral exits between the membrane-integrated Wzm subunits (Fig. 6e, g). Within the resolution limits, the core

transporter architecture comprising the NBD and TM subunits is symmetric, with similar lateral channel openings on either side of the transporter.

Due to the limited separation of the NBDs, the CBD dimer is pushed to the periphery of the NBD1 where its conserved loops straddle the hinge helix, similar to its position in the ATP-bound con-formation (Fig. 3b). Further, the close proximity of the NBDs enables direct interactions between CBD2-i and the NBD2 of the opposing Wzt subunit, likely mediated by Gln178 of the NBD2's and Arg304 of CBD2-i (Fig. 6h).

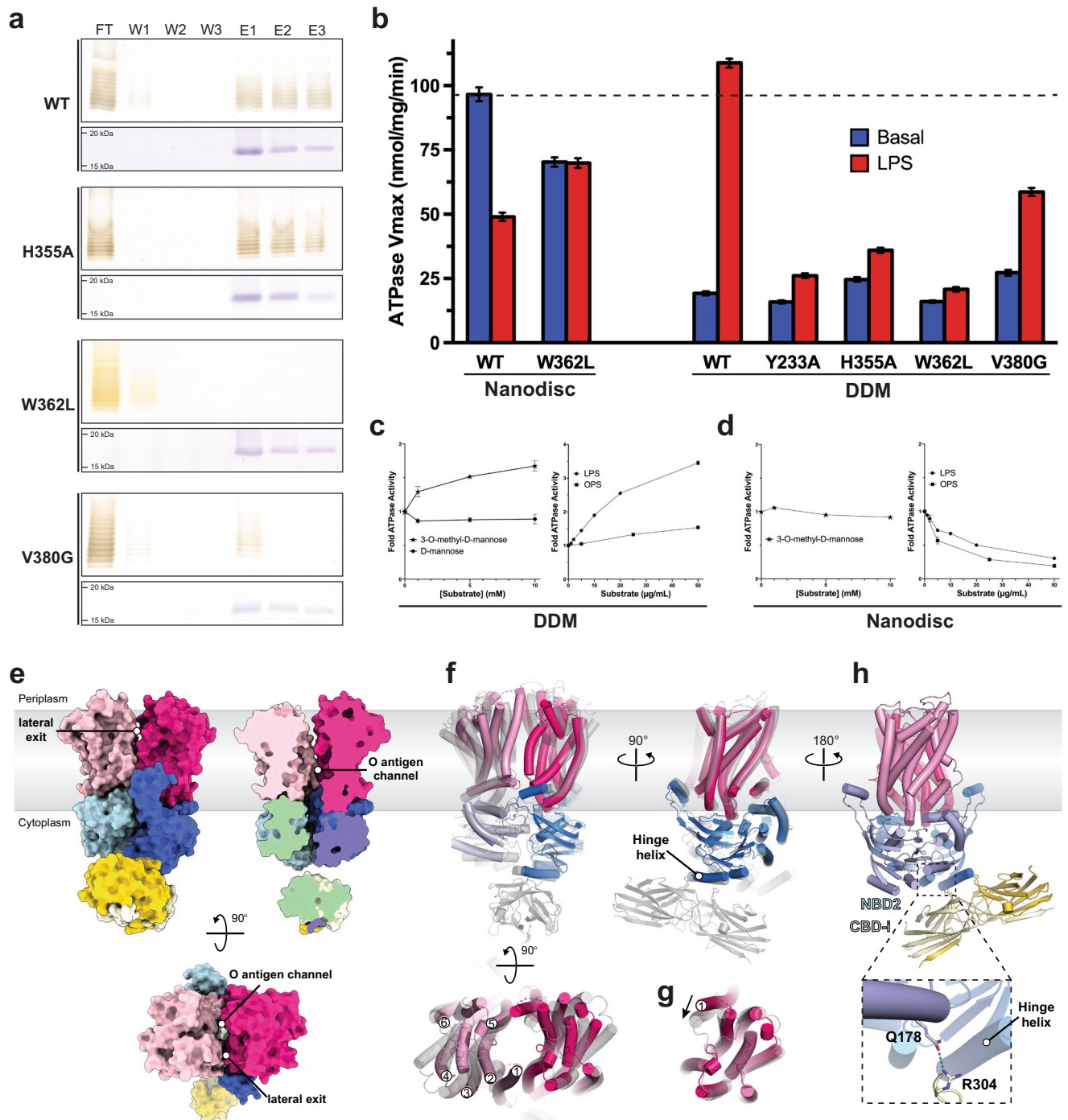

**Fig. 6 | Site-directed mutagenesis disrupts LPS binding and O antigen modulated ATPase activity. a** SDS-PAGE of pulldown experiments examining *A. aeolicus* LPS binding against different CBD mutant constructs. Top and bottom panels are silver and Coomassie-blue stained gels, respectively. Uncropped images are provided as Source Data. **b** ATP hydrolytic activity of WzmWzt and O antigen binding mutants determined in nanodiscs and DDM micelle. WzmWzt ATPase activity measuring the maximum velocity among DDM-solubilized transporter and nanodisc reconstituted constructs in the presence (LPS) or absence (Basal) of saturating *A. aeolicus* LPS. Data presents the maximum velocity measured from Michaelis Menten non-linear fit model. Each data point is the means from 3 technical repeats. Error bars represent the upper and lower 95% confidence limits from the Michaelis-Menten non-linear fit (profile likelihood) **c** WzmWzt ATPase activity in a DDM micelle titrating 3-*O*-methyl-mannose, D-mannose, LPS, and O antigen. Data points

represent averaged activities from three replicas normalized to the activity in the absence of substrate. Error bars represent the deviations from the means. **d** As in panel C but using nanodisc reconstituted WzmWzt. Individual data points are provided as Source Data. **e** Structure of WzmWzt in a DDM micelle shown as a surface colored according to the individual domains. Data points and error bars as stated under panel (**c**) **f** Superimposition of channel-forming nucleotide-free conformations of WzmWzt. The full-length transporter is shown in gray, the CBD-truncated construct (PDB: 6OIH) is colored according to its domains. **g** TM helix 1 adopts different conformations in CBD-truncated (magenta) and full-length (gray) apo WzmWzt. The structures were aligned on one Wzm subunit and the other domains are not shown for clarity. **h** Interactions between NBD2 and CBD2-i in the *in surfo* full-length WzmWzt structure.

## Discussion

The *A. aeolicus* O antigen is an alternating polymer of β-D-Rha*p*(1→2) and α-(1→3)-linked D-Rha*p* or D-Man*p* with a semi-random distribution. Identifying and biochemically characterizing the responsible glycosyltransferases will be required to delineate its biosynthesis mechanism. The identified O antigen repeat unit and 3-*O*-methyl-cap are analogous to the repeat unit structures reported from *Campylobacter fetus* serotype B and *E. coli* O8 strains, which produce D-rhamnose and D-mannose polymers, respectively[20,21].

Polyacrylamide gel electrophoresis, size exclusion chromatography, as well as NMR analyses demonstrate that *A. aeolicus* produces a high molecular weight O antigen likely exceeding 60 sugar units. Channel-forming conformations of WzmWzt suggest accommodation of about 8 sugar units inside its TM channel[15,16]; the O antigen polymer exceeds this length many times. Therefore, WzmWzt likely employs a processive, step-by-step polysaccharide translocation mechanism.

In the absence of a translocating polysaccharide and in a lipid nanodisc, WzmWzt's TM channel is closed near its periplasmic exit, both in apo and ATP bound states. This suggests that polysaccharide insertion is likely necessary to open the TM path. Channel opening results from an NBD-uncoupled rigid body movement of the Wzm subunits, as previously described[17]. We postulate that the resulting conformations are represented by transporter structures obtained in a micelle environment. During processive polysaccharide translocation, WzmWzt likely cycles through multiple NBD open and closed states, with the polysaccharide substrate spanning a continuous TM channel (Fig. 7).

The repositioning of the CBD dimer upon NBD opening was unexpected and must be functionally relevant. First, we note that the transition from the NBD-open to the -closed conformation pushes the CBD1-a protomer closer to the membrane interface, directly underneath the cytosolic channel entrance. This transition could facilitate the insertion of the substrate into WzmWzt's TM channel at the

cytosolic interface of the Wzm subunits. Second, the ATP-induced transition of CBD1-a away from the hinge helix contributing to O antigen binding likely terminates its interaction with the polysaccharide. Breaking the CBD1-a – O antigen interaction is necessary for complete substrate translocation. Third, a less pronounced NBD opening in the nucleotide-free channel-forming conformation (i.e., during polymer translocation) prevents the CBD dimer from associating with the H-loop-helix and forming a proper O antigen-binding pocket. At this time, O antigen binding by the CBD is likely unnecessary because translocation has already been initiated (Fig. 7).

Our structural insights are consistent with the CBD dimer associating with either of the NBDs, which would create indistinguishable conformations. The O antigen binding pocket is created at the CBD-NBD interface of the same Wzt protein. This suggests that both CBDs can be functional, depending on which NBD the CBD dimer is interacting with. Further, the recognition of the capping sugar together with following O antigen repeat unit elements at the CBD-NBD interface is consistent with previous complementation assays in vivo[8].

Monitoring ATP hydrolysis in the presence of the *A. aeolicus* O antigen suggests that cap recognition decreases the transporter's catalytic activity. Our structural analyses support this observation because the bound O antigen stabilizes the open conformation of the NBDs through interactions connecting Wzt1's CBD and NBD subdomains. Under physiological conditions, inhibition may be alleviated through additional interactions with the substrate's lipid-anchor, perhaps mimicked by the observed stimulatory effect of LPS *in surfo*. Further, it is conceivable that CBD - O antigen interactions precede engaging of the lipid-anchor, and reduced transporter dynamics could facilitate the latter step (Fig. 7).

Following cap binding, the reorientation of the lipid-linked O antigen likely starts with the insertion of the lipid's head group into the channel proper. Our open conformation of WzmWzt, obtained in a nanodisc, shows an asymmetric organization of the Wzm pair. On one side, the dimer accommodates a lipid molecule at the interface

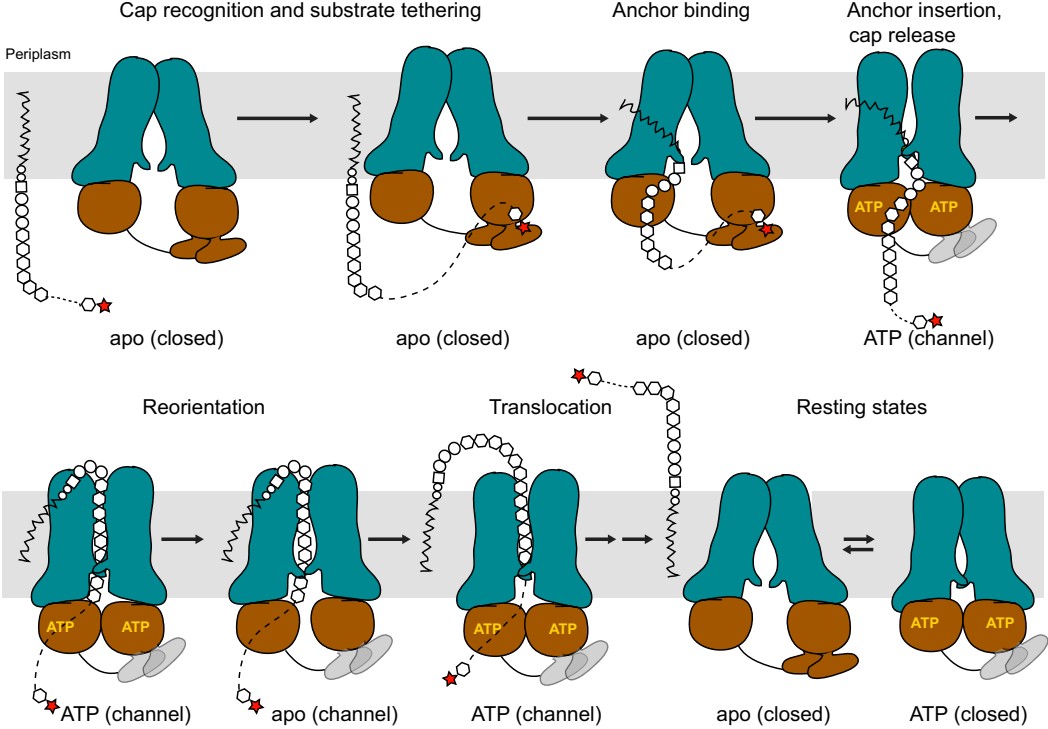

**Fig. 7 | Model of O antigen translocation.** CBD – O antigen interactions increase the local substrate concentration to initiate translocation. Cap interactions are likely disrupted in the transporter's ATP-bound conformation due to CBD

displacement (gray CBDs). The opening of the NBDs in channel-forming conformations is reduced to prevent cap re-binding. Multiple rounds of ATP hydrolysis energize polymer translocation.

between TM helices 1 and 5 from opposing Wzm subunits, right above the O antigen-binding CBD1-a (Supplementary Fig. 4d). This cytosolic portal is closed on the opposing side, suggesting that cap recognition and substrate insertion into the channel occur on the same side of the transporter. Conformational changes upon NBD closure, in particular the flipping of the LG loop towards the channel entrance, likely facilitate substrate insertion (Fig. 7).

Combined, our structural and biochemical analyses support a model in which the CBD functions to recruit the lipid-linked substrate to the transporter. O antigen-induced stalling of ATPase activity may be necessary to enable proper positioning of the lipid moiety at the channel entrance. ATP-induced NBD closure and repositioning of the CBD dimer facilitates the insertion of the lipid head group into the channel and terminates interactions with the O antigen cap. The translocating polysaccharide is accommodated in a continuous TM channel. During translocation, the separation of the NBDs is reduced to minimize CBD and O antigen interactions.

## Methods

### Lipopolysaccharide and O polysaccharide isolation and purification

*Aquifex aeolicus* biomass was purchased from Harald Huber (University of Regensburg, Germany). Cells were extensively washed in PBS buffer. LPS was extracted using the hot phenol/water method (Westphal and Jann, 1965) supplied with 2% sodium N-lauroylsacosine. The water phase was collected, and the extraction was carried out two more times with the addition of 2% N-lauroylsarcosine sodium salt in water. The water phases were dialyzed (14,000 MWCO), freeze-dried, and washed with 9:1 (v/v) ethanol in water at 4 °C. Nucleic acids and proteins were removed by 16 h treatment with 750 units of benzonase (37 °C), followed by overnight incubation with proteinase K (37 °C), respectively, followed by dialysis (14,000 MWCO) at 4 °C against several exchanges of dH$_2$O. The polysaccharide portion of the LPS (O-chain-core oligosaccharide) was liberated from the lipid A by 3 h hydrolysis with 1% HOAc, at 100 °C followed by 1h ultracentrifugation at 100,000 $g$, at 4 °C. The supernatant was extracted with chloroform (1:1, v/v) to remove traces of lipids. The freeze-dried carbohydrate fraction was resolved on a Biogel P10 column (120 ×1 cm) using 50 mM ammonium acetate as eluent. The eluting fractions were monitored on a Shimadzu RID-10 refractive index detector. The fraction enriched with the O-chain was used for structural studies.

### DOC-PAGE analysis of LPS

LPS was analyzed by PAGE using an 18% acrylamide gel with deoxycholic acid (DOC) and stained with silver, as previously described[22].

### Chemical characterization of *A. aeolicus* LPS and O polysaccharide

The glycosyl composition of LPS was determined by preparing trimethylsilyl (TMS) methyl glycosides after methanolysis (1 M HCl-MeOH at 80 °C; 18 h) in the presence of an internal standard (myo-inositol). The TMS derivatives were analyzed by GLC-MS[23,24] on a Hewlett-Packard HP5890 gas chromatograph equipped with mass selective detector 5970 MSD using an EC-1 fused silica capillary column (30 m 0.25 mm I.D.), and temperature cycle at 80 °C for 2 min, then ramp to 160 °C at 20 °C/min, and to 200 °C at 2 °C/min, followed by an increase to 250 °C at 10 °C/min with an 11 min hold. Glycosyl linkage analysis of O-chain polysaccharides was obtained by GC-MS analysis of partially methylated aldiol acetates (PMAA[25]) after 2 h hydrolysis with 2 M (v/v) TFA at 121 °C, overnight reduction with NaBD$_4$ and acetylation with acetic anhydride and pyridine. GLC-MS analysis was performed on an HP-5890 GC interfaced to a mass selective detector 5970 MSD using a Supelco SP2330 capillary column (30 0.25 mm ID, Supelco) with temperature program: 60 °C for 1 min, then ramp to

170 °C at 27.5 °C/min, and to 235 °C at 4 °C/min with 2 min hold and finally to 240 °C at 3 °C/min with 12-min hold. The substitution of neutral sugars with O-methyl groups was revealed after hydrolysis of the O-chain polymer with 2 M TFA at 121 °C for 2 h and conversion to alditol acetates[24] and analysis of the EI-MS fragments of the alditols resolved on the Supelco SP2330 capillary column using the same temperature program as for the PMAAs. Stereochemical configuration of the glycosyl residues was determined following the method of Gerwig et al.[26,27] with optically active (S)-( + )−2-butanol (Sigma).

### Determination of the average molecular weight of the *A. aeolicus* O antigen-core

The molecular weight of the liberated O antigen-core and core fractions constituting LPS was determined by using Superdex 75 10/300 GL semipreparative column interfaced with Shimadzu LC-20 type HPLC system and Shimadzu RID-10A refractive index detector. The fractions were eluted with 50 mM ammonium formate at 0.5 mL/min flow rate. The column was calibrated with commercially available standards of dextran (68,000 Da; 40,000 Da; 10,000 Da; 6,000 Da; 5000 Da), maltoheptaose (1153 Da), disaccharide (360 Da), and glucose (180 Da). In addition to the size exclusion chromatography, the molecular weight of the O antigen was also determined by NMR.

### Nuclear magnetic resonance spectroscopy of the *A. aeolicus* O antigen

To prepare the sample of the released HMW O antigen for NMR spectroscopy, the dried material (~1 mg) was dissolved in 200 μL D$_2$O (99.96% D, Cambridge Isotope Laboratories) containing 50 nmol DSS-$d_6$ (Cambridge Isotope Laboratories) and lyophilized. The deuterium exchange was repeated three more times, the dry sample was dissolved in 42 μL D$_2$O and transferred into a 1.7 mm NMR tube. NMR data were collected at 318 K on a Bruker NEO spectrometer ($^1$H, 799.71 MHz) equipped with a 1.7 mm TCI cryoprobe using standard sequences from the spectrometer library. To determine the structure of the polysaccharide, 1D $^1$H and 2D COSY, TOCSY, NOESY, HSQC, HMBC and HSQC-TOCSY spectra were collected. The $^1$H, TOCSY and NOESY spectra were collected with presaturation of the residual water signal and the HSQC spectrum was acquired with signal multiplicity editing. The mixing times were 80 ms (TOCSY), 120 ms (NOESY) and 30 ms (HSQC-TOCSY). The homonuclear correlations were collected with $^1$H spectral widths of 7143 Hz, 256 increments and 8 scans per increment. The HSQC and HMBC spectra were collected with spectral widths ($^1$H and $^{13}$C) of 9615 and 14085 Hz, 300 increments and 4 (HSQC) or 16 (HMBC) scans per increment. The NMR data were processed with Bruker Topspin (version 4.0.5) and analyzed with Topspin and CCPN Analysis (version 2.4.2[28]). $^1$H and $^{13}$C chemical shifts were referenced to the respective DSS signals at 0.0 ppm. Plots of NMR data were made in MestreNova 14.1.1.

### Wzt-CBD Expression and Purification

The *Aquifex aeolicus* VF5 Wzt-CBD (254-395) construct was cloned into a pET28a vector in frame with an N-terminal hexa-histidine tag and a Tobacco etch virus (TEV) cleavage site. The vector was transformed into C43(DE3) cells for protein expression. Expression and purification followed the same protocol as previously described[10] with additional steps to remove the His-tag. After Ni-NTA elution, the eluted protein was concentrated and dialyzed in reverse IMAC buffer (25 mM Tris pH, 50 mM NaCl, 1 mM DTT), and the His-tag was removed by TEV protease overnight at 4 °C. The protein pool was batch incubated with Ni-NTA resin, and untagged Wzt-CBD was collected in the flowthrough and wash. Wzt-CBD was further purified by size exclusion chromatography using a Superdex-200 column in 25 mM Tris pH 8.5, 50 mM NaCl, 5 mM β-mercaptoethanol or 50 mM HEPES pH 8.0 for ITC studies.

 

## LPS pulldown

Our LPS pull-down protocol was adopted and modified from a previous study (Mann et al., 2016). 0.5 mM DDM was included in all buffers for pull-down experiments examining DDM-solubilized WzmWzt constructs. 250 μg of His-TEV-Wzt-CBD, 500 μg of WzmWzt-His (DDM), 100 or 150 μg of *A. aeolicus* LPS and 250 μL of Ni-NTA resin were mixed with binding buffer (25 mM Tris pH 8.5, 50 mM NaCl, 20 mM imidazole, 1 mM DDT) in a 750 μL total volume. The CBD and *A. aeolicus* LPS were incubated on a rotor at room temperature. Ni-NTA resin was pelleted at 700 x $g$ for 2 min, and the flow-through was collected. 500 μL of binding buffer was added to Ni-NTA resin and briefly incubated on a rotor for 2 min prior to repeating centrifugation. Two additional wash steps followed. 500 μL of elution buffer (25 mM Tris pH 8.5, 50 mM NaCl, 1 M imidazole, 1 mM DDT) was added to elute the CBD. Ni-NTA resin was pelleted by centrifugation and the elution was collected. Two additional elution steps followed. Eluted CBD and LPS were examined by SDS-PAGE. CBD was visualized by Coomassie staining. LPS were treated with proteinase K and boiled prior to SDS-PAGE and visualized by silver staining. Uncropped and unprocessed images are provided in the Source Data.

## Isothermal titration calorimetry

Isothermal Titration Calorimetry experiments were carried out using a MicroCal PEAQ-ITC (Malvern) at 25 °C. 1.5 mM Wzt-CBM, 15 mM 3-*O*-methyl-D-mannose (Omicron Biochemicals) and 15 mM D-mannose (Acros Organics) were prepared in ITC buffer (50 mM HEPES pH 8.0) and loaded in the cell and injection syringe, respectively. An initial 0.2 μL injection was excluded and 75 0.5 μL injections of monosaccharide followed. The isotherm was plotted and modeled in MicroCal PEAQ-ITC Analysis Software (Malvern) and GraphPad Prism.

## CBD crystallization, data collection and structure determination

CBD crystals were prepared by mixing 1 μL of 5 mg/mL WztC with 50 mM 3-*O*-methyl-D-mannose and 1 μL of well solution (0.1 M tricine pH 8.8, 20% PEG 3350) using the hanging drop method. Drops were stored at 4 °C and crystals typically appeared after 1 day and continued to grow for a week. The crystals were soaked in cryo buffer (25% PEG 3350, 15% glycerol) prior to harvest and flash cooling in liquid nitrogen. Crystal diffraction was collected at SER-CAT (Advanced Photon Source) on the 22-ID beamline. Reflections were indexed in the P2$_1$2$_1$2$_1$ space group and integrated in XDS[29]. Intensities were scaled and merged in Aimless as part of the CCP4 software package[30]. The structure was solved by molecular replacement in Phaser[31] using PBD ID: 6O14 as a search model. Two monomers were found in one asymmetric unit. Polypeptide chains underwent iterative cycles of model building and refinement using Coot and Phenix[32,33]. 3-*O*-methyl-D-mannose was docked into the FoFc density map using LigandFit. The final model refinement generated an R$_{work}$ = 26% and R$_{free}$ = 29%. Xtriage analysis in Phenix indicated a non-origin Patterson peak of 25%, suggesting translational non-crystallographic symmetry (tNCS). Processing reflections in the P1 space group with 4 CBD dimers per crystallographic asymmetric unit also presented tNCS and similar R factors in the final model refinement.

## Cryo-EM sample preparation, data collection and processing

Wild type WzmWzt nanodiscs were purified as previously described[17]. Purified WzmWzt (*E.coli* extract lipid or POPG) nanodiscs were concentrated to 1 mg/mL and mixed with or without 1 mg/mL *A.aeolicus* O polysaccharide, 16 mM 3-*O*-methyl-D-mannose or 2.5 mM ATP. 2.5 mM MgCl$_2$ was included in the nanodisc reconstitution mixture and gel filtration buffer for WzmWzt nanodisc samples incubated with ATP. Quantifoil grids (1.2/1.3, 300 mesh) were glow discharged in the presence of amylamine. 2.5 μL of sample was applied to each grid. Grids were blotted for 4 s in 100% humidity and plunged frozen in liquid ethane. Cryo-EM images were collected using EPU on a Titan Krios

equipped with a K3/GIF detector (Gatan) at the Molecular Electron Microscopy Core (University of Virginia School of Medicine).

Movies were imported into cryoSPARC v3.0 followed by patch motion correction and patch contrast transfer function estimation. Exposures of good resolution quality were curated for further processing. Initial particles were automatically selected by Blob picker (cryoSPARC) using a 20 Å lowpass filter. These particles were then extracted at a box size of 256 pixels. Particles were then used to generate ab initio models and subsequently sorted by iterative cycles of 2D and 3D classification. Selected 2D templates were used for template-based particle picking, followed by 2D and 3D sorting. The final volumes were refined using either non-uniform or local refinement (cryoSPARC) to generate high-resolution volumes (3.3–4 Å). Additional details can be found in the Supplementary Figs. 3, 5 and 6.

## Model building

The nucleotide-free full-length WzmWzt structure was generated by rigid body docking in Chimera[34] and Coot of the individual subunits into the cryo EM map using PDB ID: 7K2T as template. The obtained model was completed and refined in Coot. The linker regions connecting the NBD and CBD sub-domains of Wzt were built de novo in Coot. The model was refined in Phenix.refine[32]. The WzmWzt structure bound to ADP:Mg$^{2+}$ was indistinguishable from the apo conformation and was used to generate the final model due to superior EM map quality. For the 3-*O*-methyl-D-mannose bound complex, 3-*O*-methyl-D-mannose was modeled into density peaks according to the high-resolution CBD-cap crystal structure. The model was refined in Phenix and manually adjusted in Coot. For the O antigen bound complexes, a trisaccharide representing the first three D-rhamnose units from the non-reducing end [α-D-Rha3Me(1 → 3)-β-D-Rha(1 → 2)-α-D-Rha] of the *A. aeolicus* O antigen was built using the GLYCAM server (https://glycam.org/). The capped non-reducing end of the oligosaccharide was oriented and docked in the same space where 3-*O*-Me-Man bound in the cap-CBD crystal structure. The oligosaccharide model was manually refined into the coulombic potential map.

## ATPase activity assays

ATPase activity was quantified using an enzyme-coupled assay as previously described[16]. DDM-solubilized and nanodisc-reconstituted WzmWzt were purified by gel filtration, and the peak fraction (0.5–1.3 mg/mL) was collected to test for activity. Reaction samples were pre-incubated at 27 °C for 30 min. ATPase activity was initiated upon the addition of ATP, and the depletion of NADH was monitored at 340 nm for 1 h. Substrate-titration assays were supplied with 1 mM ATP. Data was processed in GraphPad Prism. The rate of NADH depletion was converted to nmols of ATP hydrolyzed using an ADP standardized plot. All experiments were performed with triplicate technical repeats.

## Statistics and reproducability

Gel images presenting LPS pulldowns from Fig. 6a and Supplementary Fig. 7a are representative results from two independent experiments except for WzmWzt H355A and V380G, which were performed once.

## Reporting summary

Further information on research design is available in the Nature Research Reporting Summary linked to this article.

## Data availability

Atomic coordinates and structure factors of 3-*O*-Me-Man-bound Wzt-CBD have been deposited in the Protein Data Bank under the accession code 8DKY (Supplementary Table 4). Atomic coordinates and cryo-EM maps of WzmWzt have been deposited in the Protein Data Bank and Electron Microscopy Data Bank under accession codes 8DOU and

EMD-27623 (ADP-bound), 8DKU and EMD-27491 (O antigen-bound), 8DN8 and EMD-27556 (3-*O*-Me-Man-bound), 8DNC and EMD-27563 (O antigen and ADP-bound), 8DNE and EMD-27564 (ATP-bound), and 8DL0 and EMD-27494 (nucleotide-free, DDM-solubilized) (Supplementary Table 5). Integrated heat injections and Michaelis-Menten data are provided in the Source Data file. Source data are provided with this paper.

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

## Acknowledgements

We thank K. Dryden and M. Purdy from the Molecular Electron Microscopy Center at the University of Virginia for support and are indebted to Ruoya Ho for help with cryo-EM data collection. We also thank the National Cryo-EM Facility (NCI) for collecting cryo-EM data for the ADP-bound WzmWzt. J.Z. and N.S were supported by NIH grants R01129666 and R35GM144130. I.G. is a recipient of the Boehringer Ingelheim Fellowship. Work at the Center for Plant and Microbial Complex Carbohydrates at the Complex Carbohydrate Research Center was supported by the US Department of Energy (DOE), Office of Science, Basic Energy Sciences (BES), under Award DE-SC0015662. J.Z. is an investigator of the

Howard Hughes Medical Institute. This article is subject to HHMI's Open Access to Publications policy. HHMI lab heads have previously granted a nonexclusive CC BY 4.0 license to the public and a sublicensable license to HHMI in their research articles. Pursuant to those licenses, the author-accepted manuscript of this article can be made freely available under a CC BY 4.0 license immediately upon publication. X-ray data were collected at Southeast Regional Collaborative Access Team (SER-CAT) 22-ID (or 22-BM) beamline at the Advanced Photon Source, Argonne National Laboratory. SER-CAT is supported by its member institutions, and equipment grants (S10_RR25528, S10_RR028976 and S10_OD027000) from the National Institutes of Health.

## Author contributions

J.Z. and N.S. designed the experiments. I.G. solved apo and ADP-bound structures of WzmWzt. A.M. and J.V. purified and characterized the A. aeolicus O antigen. N.S. determined the cap-bound CBD crystal structure, performed all mutagenesis experiments, and determined cap-bound, O antigen-bound and DDM-solubilized WzmWzt structures. N.S., A.M., I.G., J.V., D.H., P.A. and J.Z. evaluated and interpreted the data. J.Z. and N.S. wrote the first manuscript and all authors edited it.

## Competing interests

The authors declare no competing interests.
