## [Peer Review File · Nature Communications]

Molecular basis for polysaccharide recognition and modulated ATP hydrolysis by the O antigen ABC transporterREVIEWERS' COMMENTS

Reviewer #1 (Remarks to the Author):

In this study, Spellmon et al. use a combination of biochemistry and structural biology to elucidate the chemical structure of the O antigen cap from *A. aeolicus* and investigate how this cap is recognized by the O antigen transporter, WzmWzt. These studies have led to insights into how the O antigen cap binds to the carbohydrate binding domain of Wzt and how this binding is coordinated with ATP hydrolysis for substrate transport.

Overall, the study by Spellmon et al. is well executed and provides interesting insights to the mechanism of a unique ABC transport system. The structural data are of high quality and support the claims made by the authors. I had a few questions and suggestions that may help in revisions. In particular, several sections would benefit from an initial statement of the hypothesis being tested or the motivation for the experiments to follow, and also a conclusion at the end. Currently, the reader is sometimes left to piece this together on their own, and it took some detective work on the reader's part to figure things out (I have tried to note a few examples of this below). But scientifically I have no major criticisms, and I think this is a very nice piece of work.

Minor Comments:

Lines 203-205: It would be helpful to overlay the substrate bound CBD structure with the apo structure when comparing the two, to show the reader the movement of the trap loop.

Lines 214-219: Refers to apo and ADP-bound structures, but Fig 3a and ExtData 3 only shows ADP-bound. I got a little confused in this paragraph about whether the apo structure was previous work or a new structure presented here. The first column of Sup Table 4 seems to indicate data collected with a 3.65 Å map but no model. I think it would be helpful for the authors to better explain what is going on here (e.g., were the maps for apo and ADP in nanodiscs basically identical, and therefore only the ADP structure is shown?). I think that would help to avoid confusion.

Lines 221-223: I think it could be helpful to the reader to show this in figures somehow (previous ATP bound structure vs new ADP-bound structure).

Lines 226-230: Hinge-helix and H-helix are easily confused. Could the name of one be changed to make them more obviously different?

Lines 235-237: "This transition moves the CBD sub-domain connected to the NBD the CBD dimer is interacting with (referred to as CBD-a, 'a':active) away from the membrane": I wonder if it could be stated more clearly? Could the authors instead define that NBD1 and CBD1 are from the same polypeptide (and NBD2 and CBD2 are one polypeptide), and if the CBD dimer is bound to NBD1, then CBD1 moves away from the membrane? I think that would be more clear, but I appreciate that this is a complicated idea to try to convey!

Lines 230, 235, and 241: "Fig. 3a": should this be referring to the blowup in Fig. 3b? Would it make sense to also show the interactions made by this loop in the ATP-bound state, so the reader can see the differences indicated in the text?

Line 271: the "LG-loop" is labeled "PG-loop" in the figure (3c)- is this a mistake?

Lines 281-321: By putting together the section title and the contacts made by Tyr233, I gather that the best explanation for binding to one CBD site and not the other is Tyr233 (which is only present at the "a" site). If that is indeed the model, I suggest the authors say that explicitly (with the appropriate degree of qualification) so the point is clear.

Fig. 3a: Is the box with a small "C" in it meant to indicate that this region is blown-up in panel C? Mention in figure legend? Use dotted lines to indicate enlargement as in panel b?

Figure 3c: What does each color represent (e.g., what is cyan)? Please state in legend

Line 342: "CBM-a": should this be CBD-a?

Line 379: Should be "reports" (not report)?

Line 370-401: I think it would make this more clear if this section started with a hypothesis/rationale for the experiments that follow. Is this more about whether O antigen stimulates vs inhibits activity? Or is ATPase activity just being used as a proxy for binding to the intact transporter? Presumably the inhibition of activity in nanodiscs is more physiological, yet most of the data for mutants is in DDM,

where O antigen stimulates activity. And in general, one normally expects substrate to stimulate the ATPase activity of an ABC transporter. So at the end of this section, I am a little uncertain what to conclude, or what hypothesis/model was being tested.

Lines 393-394: It would be helpful if the authors provided their interpretation of the finding that the hydrolytic rate of the mutant is reduced by ~25% relative to WT? Is that to say that this data point is unreliable and should be ignored? Then why not remove it from the manuscript?

In 404-405: This first sentence leads the reader to think this structure will shed light on the higher LPS-stimulated ATPase activity in DDM, but unless I missed it, no explanation is given?

Sup. Table 4: The authors should include statistics that speak to the agreement between the model and their experimental map (such as CCmask, CCpeaks, CCvolume, and CCbox).

Fig. 4 legend: change "electron potential" to "coulombic potential"

Fig. 7 (model): One thing that seems surprising to me about the repositioning of the CBDs is that either they must unbind from the hinge helix and then rebind to the H-helix, OR they remain associated but somehow "slide" across the surface of the NBD. If the CBDs slide, they would remain associated with the same NBD, but if they unbind, they could potentially jump to the other NBD. Is it worth discussing how this might work? Discuss other gaps in knowledge in discussion?

ExtData 7a: The changes here are not very obvious. Can this be quantified in some way and normalized to the amount of transporter pulled down? If not, I would probably remove it from the manuscript.

Reviewer #2 (Remarks to the Author):

Despite the wealth of mechanistic details on the pathway of O antigen synthesis to export to the OM, there are still details yet to be elucidated. The WzmWzt ABC exporter, responsible for the recognition and transport of capped O antigen transport across the inner membrane, utilizes CBD, but the nature of the caps in *A. aeolicus*, details of the binding and recognition mechanism are unclear. This manuscript by Spellmon et al. provides a more complete model for transport, providing additional conformations of WzmWzt in variable membrane/ micelle environments

Using a combination of MS and NMR to identify composition and methylation native *A. aeolicus* O antigen polymer caps reveals a combination of D-Rhap and D-Manp polymers.

This series of experiments was followed up by binding assays with the Wzt's binding domain, which binds 3-O-methyl-D-mannose but not D-mannose, and the crystallographic structure of the CBD-3-O-methyl-D-mannose complex, indicating the hydrophobic path and trap loop necessary for O-antigen binding. The full-length transporter in nanodisks has reduced activity in the presence of polysaccharide, decreasing the ATP hydrolysis of the transporter

The cryo-EM WzmWzt structure in nanodisk of the apo, ADP, 3-O-Me-Man coupled with the 4-angstrom resolution structure in detergent revealed a rearrangement of the nucleotide binding domain and c-terminal binding domain, leading to a model, partially supported by these structures, and previously published conformational states.

The manuscript could be improved with additional information on the resolution in areas of note, especially polysaccharide interaction domains. While the model is feasible, clearly linking structure (PDB ID) to states in the model would be helpful. The biochemistry in this manuscript is strong adding to the excitement of the model. The methodology used to obtain the structures and activity assays presented in this manuscript is sound and well established in the field; the manuscript and follow-up references provide details that would allow for reproducibility.

Reviewer Comments

Reviewer #1 (Remarks to the Author)

Overall, the study by Spellmon et al. is well executed and provides interesting insights to the mechanism of a unique ABC transport system. The structural data are of high quality and support the claims made by the authors. I had a few questions and suggestions that may help in revisions. In particular, several sections would benefit from an initial statement of the hypothesis being tested or the motivation for the experiments to follow, and also a conclusion at the end. Currently, the reader is sometimes left to piece this together on their own, and it took some detective work on the reader's part to figure things out (I have tried to note a few examples of this below). But scientifically I have no major criticisms, and I think this is a very nice piece of work.

We thank Reviewer #1 for the thoughtful feedback and the minor points raised.

Minor Comments:

Lines 203-205: It would be helpful to overlay the substrate bound CBD structure with the apo structure when comparing the two, to show the reader the movement of the trap loop.

>>We added an arrow to Fig. 2e indicating the trap-loop movement from the apo to the cap-bound state. We hope that this clarifies the conformational changes.

Lines 214-219: Refers to apo and ADP-bound structures, but Fig 3a and ExtData 3 only shows ADP-bound. I got a little confused in this paragraph about whether the apo structure was previous work or a new structure presented here. The first column of Sup Table 4 seems to indicate data collected with a 3.65 Å map but no model. I think it would be helpful for the authors to better explain what is going on here (e.g., were the maps for apo and ADP in nanodiscs basically identical, and therefore only the ADP structure is shown?). I think that would help to avoid confusion.

>>We introduced the following sentence in lines 237-239 to clarify that the apo and ADP-bound states are essentially identical: "The overall conformation of the transporter in the apo and ADP-bound states is essentially the same, so we selected the ADP-bound model for analysis".

Lines 221-223: I think it could be helpful to the reader to show this in figures somehow (previous ATP bound structure vs new ADP-bound structure).

>> Figure 3B shows an overlay of the new NBD-open and previously published ATP-bound states.

Lines 226-230: Hinge-helix and H-helix are easily confused. Could the name of one be changed to make them more obviously different?

>>We have replaced H-helix with H-loop-helix to avoid confusion.

Lines 235-237: "This transition moves the CBD sub-domain connected to the NBD the CBD dimer is interacting with (referred to as CBD-a, 'a':active) away from the membrane": I wonder if it could be stated more clearly? Could the authors instead define that NDB1 and CBD1 are from the same polypeptide (and NBD2 and CBD2 are one polypeptide), and if the CBD dimer is bound to NBD1, then CBD1 moves away from the membrane? I think that would be more clear, but I appreciate that this is a complicated idea to try to convey!

>> This paragraph has been revised. We now distinguish the Wzt subunits of the transporter as Wzt1 and Wzt2 and the corresponding subdomains as CBD1/2 and NBD1/2 to simplify the description.

Lines 230, 235, and 241: "Fig. 3a": should this be referring to the blowup in Fig. 3b? Would it make sense to also show the interactions made by this loop in the ATP-bound state, so the reader can see the differences indicated in the text?

>>We introduced another panel to SI Fig. 4 showing the specific interactions of the CBD dimer with the NBD's Hinge-helix in the ATP-bound conformation.

Line 271: the "LG-loop" is labeled "PG-loop" in the figure (3c)- is this a mistake?

>> Thank you for pointing out the mistake. It has been corrected.

Lines 281-321: By putting together the section title and the contacts made by Tyr233, I gather that the best explanation for binding to one CBD site and not the other is Tyr233 (which is only present at the "a" site). If that is indeed the model, I suggest the authors say that explicitly (with the appropriate degree of qualification) so the point is clear.

>> That is correct. We included a short paragraph to stress this point. "The side chain of NBD1 Tyr233 appears to be key for contacting the O antigen. The arrangement of the CBD dimer and NBD1 appears necessary for Tyr233 to contact a single O antigen chain. CBD2-i likely does not possess an O antigen since its O antigen binding pocket is segregated away from Tyr233 of NBD2."

Fig. 3a: Is the box with a small "C" in it meant to indicate that this region is blown-up in panel C? Mention in figure legend? Use dotted lines to indicate enlargement as in panel b?

>>We included: "Boxed region is blown-up in (c)" in the legend of Figure 3. We would prefer omitting dotted lines as lines clutter the overall figure.

Figure 3c: What does each color represent (e.g., what is cyan)? Please state in legend

>>We now specify "Cyan cartoon represents the IF-TM1 connection observed in the ATP-bound WzmWzt conformation (PDB ID: 7K2T[<https://www.rcsb.org/structure/7K2T>])" in the legend of Figure 3.

Line 342: "CBM-a": should this be CBD-a?

>> Thank you for pointing out the mistake. It has been corrected

Line 379: Should be "reports" (not report)?

>> Thank you for pointing out the mistake. It has been corrected

Line 370-401: I think it would make this more clear if this section started with a hypothesis/rationale for the experiments that follow. Is this more about whether O antigen stimulates vs inhibits activity? Or is ATPase activity just being used as a proxy for binding to the intact transporter? Presumably the inhibition of activity in nanodiscs is more physiological, yet most of the data for mutants is in DDM, where O antigen stimulates activity. And in general, one normally expects substrate to stimulate the ATPase activity of an ABC transporter. So at the end of this section, I am a little uncertain what to conclude, or what hypothesis/model was being tested.

>> Indeed, we utilized ATPase stimulation as a proxy tool to gauge O antigen cap binding among different mutant constructs, and we demonstrate cap-binding does influence the ATPase activity of the transporter in DDM and nanodisc conditions. To be clear, we believe our present data is insufficient to explain how LPS significantly stimulates the ATPase activity of WzmWzt in a detergent micelle, and we would prefer to refrain from interpreting ATPase stimulation into our translocation model due to possible artificial causes from the transporter-detergent micelle. In a more physiological setting (nanodisc), cap-binding hinders ATPase activity, and we believe this may stabilize the transporter in the open-NBD conformation. This stabilized and ATPase-inhibited conformation may assist contacting the undecaprenyl-pyrophosphate portion of the substrate towards the TM1/TM5 cytosolic gap in the Wzm dimer. Once the O antigen is loaded into the TMD channel, the CBD is displaced in the open channel conformation to release the capped O antigen (Figure 6e-h). It's possible an O antigen-loaded transporter could enhance ATPase activity as observed by other ABC transporters, but further studies are required in a physiological environment.

Lines 393-394: It would be helpful if the authors provided their interpretation of the finding that the hydrolytic rate of the mutant is reduced by ~25% relative to WT? Is that to say that this data point is unreliable and should be ignored? Then why not remove it from the manuscript?

>> It is unclear why the W362L mutant exhibits reduced catalytic activity. Many factors, including suboptimal nanodisc reconstitution could contribute to it. Therefore, we deleted the sentence from the manuscript.

In 404-405: This first sentence leads the reader to think this structure will shed light on the higher LPS-stimulated ATPase activity in DDM, but unless I missed it, no explanation is given?

>> It is possible that the in surfo channel forming conformation of WzmWzt represents the conformation in which the transporter exhibits reduced ATPase activity. Addition of LPS could modulate micelle properties such that a conformation similar to the nanodisc reconstituted states is stabilized (with higher ATPase activity). We tried to test this hypothesis by determining in surfo structures with and without LPS. Unfortunately, we have not been able to prepare cryo grids at a quality sufficient for structure determination, primarily due to high background/thick ice and particle aggregation. Therefore, we cannot provide a structural rationale for the observed differences in response to LPS binding by WzmWzt.

Sup. Table 4: The authors should include statistics that speak to the agreement between the model and their experimental map (such as CCmask, CCpeaks, CCvolume, and CCbox).

>> CC scores are now included in Supplementary Table 5.

Fig. 4 legend: change "electron potential" to "coulombic potential"

>> Corrected in Figure 4 (line 701) and methods (line 878)

Fig. 7 (model): One thing that seems surprising to me about the repositioning of the CBDs is that either they must unbind from the hinge helix and then rebind to the H-helix, OR they remain associated but somehow "slide" across the surface of the NBD. If the CBDs slide, they would remain associated with the same NBD, but if they unbind, they could potentially jump to the other NBD. Is it worth discussing how this might work? Discuss other gaps in knowledge in discussion?

>> We believe the CBD slides along the same NBD (i.e. NBD1) based on our observation of the "intermediate" volume in which the CBD and NBD1 arrange together in between the ADP and ATP-bound states. We were unable to resolve the intermediate volume to high resolution due to particle heterogeneity and low particle count, but the movement of the CBD from apo to ATP-bound can be better observed in a 3D variability display (not shown here). While we can't completely rule out the possibility of the CBD jumping off and attaching to NBD2 between apo and ATP-bound states, we believe sliding is most likely.

ExtData 7a: The changes here are not very obvious. Can this be quantified in some way and normalized to the amount of transporter pulled down? If not, I would probably remove it from the manuscript.

>> Supplementary Figure 7 ultimately demonstrates that Aquifex aeolicus LPS and lipid A will naturally embed into the DDM micelle of WzmWzt. We don't think careful quantification is necessary since we're not making conclusions regarding cap-binding among the full-length transporter constructs.

Reviewer #2 (Remarks to the Author)

The manuscript could be improved with additional information on the resolution in areas of note, especially polysaccharide interaction domains. While the model is feasible, clearly linking structure (PDB ID) to states in the model would be helpful. The biochemistry in this manuscript is strong adding to the excitement of the model. The methodology used to obtain the structures and activity assays presented in this manuscript is sound and well established in the field; the manuscript and follow-up references provide details that would allow for reproducibility.

>> We thank Reviewer #2 for carefully reviewing our manuscript. We incorporated the reviewer's suggestions whenever possible.